# Food Next Door: From Food Literacy to Citizenship on a College Campus

**DOI:** 10.3390/ijerph18020534

**Published:** 2021-01-11

**Authors:** Nanna Meyer, Mary Ann Kluge, Sean Svette, Alyssa Shrader, Andrea Vanderwoude, Bethany Frieler

**Affiliations:** 1Department of Human Physiology and Nutrition, Johnson E. Beth-El College of Nursing and Health Sciences, University of Colorado Colorado Springs, Colorado Springs, CO 80918, USA; ssvette@uccs.edu (S.S.); ashrader@uccs.edu (A.S.); avanderw@uccs.edu (A.V.); bfrieler@uccs.edu (B.F.); 2Dining and Hospitality Services, University of Colorado Colorado Springs, Colorado Springs, CO 80918, USA; 3Department of Health Sciences, Johnson E. Beth-El College of Nursing and Health Sciences, University of Colorado Colorado Springs, Colorado Springs, CO 80918, USA; mkluge@uccs.edu

**Keywords:** food literacy, food citizenship, local food, food value chains, experiential learning, nutrition education, transformation

## Abstract

Industrial agriculture and food corporations have produced an abundance of food that is highly processed, nutritionally poor, and environmentally burdensome. As part of a healthy campus initiative, generated to address these and other food production and consumption dilemmas, a student-run “local and sustainable” food establishment called Food Next Door (FND) was created. This intrinsic case study evaluated food literacy in health science students, faculty, and staff first as a pilot to build the case for FND and further explicated customers’, volunteers’, and leads’ experiences with FND, identifying potential pathways from food literacy to citizenship. Ten returning customers, eight recurring nutrition student volunteers, and three graduate student leads participated in interviews that were analyzed for themes and subthemes. The findings show a progression in themes. Customers’ experiences highlight FND’s fresh, flavorful food, smiling and supportive staff, and personal transformation. Volunteers’ themes identified greater awareness of new foods and plant-based eating, acquiring new knowledge and skills in commercial kitchens, and deepening their connection to food, each other, and to where food comes from. Leads’ themes show opportunities to gain managerial skills, a deeper understanding of food and skills from being immersed in value-based food systems, and confidence in peer teaching. Experiencing and becoming part of the food value chain through FND built food literacy, shifted values, and transformed students into food citizens.

## 1. Introduction

In the last decade, attention to integrate sustainability in nutrition has increased. A sustainable diet was first defined by the Food and Agriculture Organization based on a trans-disciplinary workshop among professionals in environmental and nutritional sciences [1]. Subsequently, there have been numerous calls to nutrition organizations to include sustainability considerations in food-based dietary guidelines [2,3]. Most recently, the EAT-Lancet Commission on Food, Planet, and Health published a critical review and call to action for the “Great Food Transformation” through healthy eating from sustainable food systems [4].

Several practice and concept papers have also been published recently to raise awareness in nutrition practitioners and organizations for curricular and practice integration of sustainable food systems topics and experiences around the globe [5,6,7,8,9,10,11,12,13]. Recently, Ingram and colleagues published a framework for post-graduate education for food systems professionals, which should include dietitians, characterizing food systems knowledge, skills, and values as critical components for their inter-disciplinary program [14]. In the United States (US), the Academy of Nutrition and Dietetics (AND) published a framework to illustrate entry points for sustainable food systems integration in nutrition professionals addressing education and training from research to practice and policy [9]. This most recent publication is built on various previous practice papers, including Standards of Practice for Professional Performance in Sustainable, Resilient, Healthy Food and Water Systems [8]. While there is a good body of literature targeting US registered dietitians (RDs), there are only three out of 38 competencies and three out 220 performance indicators that address sustainable food systems and/or agriculture in curricular guidelines for Future Graduate Programs in nutrition and dietetics [15]. Globally, the International Confederation of Dietetics Professions has recently published a toolbox for sustainability integration into dietetics practice, which includes both self-paced, directed, or free flow learning and a comprehensive resource list [11]. There are currently no globally recognized competencies for nutrition professionals in sustainable food systems; however, the aforementioned examples [5,6,7,8,9,10,11,12,13,14,15] indicate there is some momentum for future inclusion.

Food literacy [16,17,18,19] and food citizenship [20,21] are both well defined, but these concepts have mostly been related to the public [17,18,19,21] and not the professional giving food and nutrition advice, since it is assumed that the professional has high food literacy [16]. Additionally, the two concepts of food literacy and citizenship, while connected initially by Schnögl et al. [22], have only recently been linked in the literature conceptually, and in an academic setting [23], but not in the context of dietetics students with career paths of becoming RDs.

It is generally accepted that food literacy is focused on increasing food and nutrition knowledge and practical skills in food selection, planning, and preparation that lead to healthful eating behaviors. Food literacy also focuses on the interactions among individuals and sharing of information and actions with regard to food and eating, for example from expert to client. However, food literacy also encompasses a broader perspective with critical reflection and integration of food’s social and environmental contexts, including the sustainability of food production [17,18,19]. It is in this latter part of food literacy’s deeper discourse in food systems with impacts on social and environmental issues [18,23] where food literacy is thought to cross into food citizenship [23]. Food citizenship is concerned with engagement, decision making, and advocacy toward a more democratic, socially and economically just, and environmentally sustainable food system [20], but more needs to be understood about what type of learning experiences ignite food citizenship.

Most recently, Vettori et al. evaluated food (and nutrition) literacy in terms of antecedents, components, and consequences [19]. One of the antecedents of food literacy listed is related to agriculture and closer connections to food producers. As consumers are drifting further apart from nature and the places where food is grown, food literacy and relationships to food, culture, and the land also vanish. While there is great interest in food and diets, these trends are mostly related to instrumental values of food, such as fuel for exercise or high-protein diets for weight loss, which are typically disconnected from social or environmental values [24]. However, with the COVID-19 pandemic, the global food supply chain experienced variable stability [25], exposing many of its weak links and re-igniting alternative food systems such as food value chains.

Food value chains are alternative arrangements most often known as regional or local food systems. Value chains transform the traditional competitive seller/buyer relationships into a collaborative approach. Transparency, working together, and providing fair shares to all partners under unified environmental or social values are characteristics of food value chains [26]. Food value chains are reappearing as alternative systems but remain less known as learning grounds for students, even though food production and supply chains per se are critical components of food systems knowledge [6,7,9]. There is often a lack of detail in distinguishing agricultural scales, methods, food quality, and shared values as well as the degree of student involvement and possible progression to optimize learning.

One of the caveats regarding the integration of sustainability into the curriculum for dietitians is that the field of sustainability sciences and practices, including food systems work, is very broad and complex, and thus, it easily gets marginalized in an already packed science-focused nutrition and dietetics curriculum. This is in contrast to the urgency for action to promote sustainable, healthy food systems, where the nutrition professional sits at the center of this work [9] to promote change on individual, community, and institutional levels.

Universities, and specifically dining halls, have been recognized as possible living learning labs for food literacy for students [27]. Campus farms and gardens have received some attention in supplementing students’ classroom learning in agroecology [28] and other fields [29]. Likewise, studies have evaluated academic courses [30,31,32] and their impact on eating behavior and co-curricular activities [33] using experiential learning with a focus on cooking classes. However, there are currently no studies in which nutrition students—future RDs—experientially learn to integrate healthy eating from sustainable food systems through food value chains.

Thus, more needs to be known about educational opportunities that promote learning engagements in sustainable foods systems, especially in cyclic, value-based food systems where students, and in this case, nutrition majors, partake in all aspects from food production to consumption. Positioning such experiences for students and future professionals as food literacy and citizenship also remains unexplored. Therefore, the aim of this study was to describe experiences in university customers eating at, and students volunteering for and leading Food Next Door (FND)—a student-run, local food establishment—on a university campus. A secondary aim was to identify potential pathways from food literacy to citizenship through experiences with FND.

## 2. Materials and Methods 

### 2.1. Situating the Study and Case Description

When the University of Colorado Colorado Springs (UCCS) foodservice transitioned to self-operation away from a corporate management company, one of the main reasons was to create academic connections through food systems-based, living–learning lab opportunities on and off campus. At this time, a program called SWELL (Sustainability, Wellness & Learning, see www.uccs.edu/swell) was launched to promote wellness through hands-on learning and skill—building in sustainability practices to regenerate human health, cultivate a mindful society, and protect the environment.

Part of SWELL was the creation of a local food retail venue in coordination with UCCS’ Dining and Hospitality Services (DHS) called FND. Sourcing local, sustainable food from Colorado family farmers, complemented with food grown on campus, was one of the goals of FND. FND produced the so-called SWELL meals, which were mostly plant-based, with a protein flip option [34]—the “SWELL burger” (with less but better meat; see Figure 1)**,** both menu options aiming for what SWELL stands for—“good for you and good for the planet”. The UCCS Farmhouse, also created at this time, helped FND access zero-mile produce from the campus farm, complementing procurement through Southern Colorado’s farms. FND was a fully student-run enterprise aligned with a student-developed sourcing approach (Figure 1) and circular process, which required planning menu design and procurement from nearby farms, working on the campus farm to harvest produce, cooking in commercial kitchens with volunteers, serving with food literacy and taste education and eye-catching beauty, and recovering, reusing, or composting before doing it all over again the following week (Figure 2).

FND was supervised by faculty, directed by graduate assistant leads (SWELL students), and supported by undergraduate volunteers (mainly nutrition students). SWELL meals and burgers were served first as a pilot in to-go containers in 2015/2016, second in a market-style retail dining operation in 2016/2017, and finally in residential locations 2017/2018, two to three times per week. FND customers were campus and community-based patrons. Further, volunteering for, and leading FND was expected to boost food literacy and help rebuild campus culture and a sense of place through the creation of a different food experience on campus. Graduate students were engaged in FND as leads. FND required them to handle heavy workloads, with a focus on food supply issues, bridging gaps from on-campus to rural agriculture and academics to auxiliary services, to name a few. FND was also tasked with refreshing the messaging around healthy food, engaging eaters with the ideas of digital-free community lunches at designated tables, and instilling the values of locally grown food. Volunteer students were primarily in their last year before graduation and signed up in practicum rotations with the goal to deepen their learning through various self-selected dietetics rotations. Ethnographically, FND’s culture is best described as a student-led, multi-disciplinary, experiential process that is embedded in academics, implemented through the collaboration with dining services, and enabled and elevated through the local and regional food system and its farmers.

FND was not the first food systems initiative integrating nutrition undergraduate and graduate students at UCCS. Our work started in 2009 with the first students involved in food and farm field trips as part of community nutrition coursework in dietetics. This developed relationships with local farmers, food hubs, and chefs. In the following years, community funding led to graduate student assistantships on farms with the vision to create mutual benefits between the dietitian and the farmer working together: the farmer teaches the dietitian where food comes from, how it is grown and from which seeds, when it is harvested, how it is priced and sold, and its history and cultural significance. At the same time, the dietitian educates the farmer in nutrition literacy and establishes the connection to the consumer as a health professional promoting fresh, locally grown food using gained knowledge and skills in food literacy from the farmer. This collaboration led to the Flying Carrot Food Literacy project with graduate students working on farms and becoming the link to consumers at a local farmers market. As a result of its transformative significance, this work also initiated the first course in sustainable food systems within nursing and health sciences departments. The course “Food, Culture, Community, and Health” became a core course for graduate students in sport nutrition and a sustainability-flagged course for the UCCS’ Compass Curriculum. These past occurrences are mentioned here to position FND, since FND grew out of the work with the Flying Carrot Food Literacy project, which focused on the community. UCCS’ foodservice transition to self-operation in 2014, the campus farm and farmhouse, and the built relationships with the urban and rural farming community, including food hubs, enabled FND. Finally, FND gave rise to other academic and community programs, one of which became the UCCS Grain School—an academic and community course with a focus on heritage grains and the grain value chain with the goal of re-establishing a heritage grain economy in Colorado. For a summary of the UCCS food journey, see [35]. 

### 2.2. Theoretical Underpinnings: Diffusion of Innovation 

Diffusion of innovation (DOI) theory focuses on how new ideas and practices are adopted over time [36,37]. DOI provides insight into how communication about the characteristics of an innovation influences adoption and how innovations spread through a social system. Five stages of the innovation decision-making process are identified as the following: awareness and knowledge of the innovation, persuasion toward the innovation (advantages, expected outcomes), the decision to act (accepting or rejecting the innovation), implementation (adoption and trial), and confirmation (commitment to use, continue, or discontinue adoption) [37]. DOI precepts include identifying channels of communication, formal and informal, as well as soliciting subjective evaluation from those trying the innovation for the first time (considered credible resources of information). DOI helped us explore the FND experience through multiple channels and responses to our innovation. With roots in social change [36], DOI aligned theoretically with our research aim, as it concerns itself with social justice, empowerment, and healthful environments, not just individual behavior change.

In qualitative research (QLR), the use of theory varies extensively [38]. For this study, we did not determine diffusion rate, the rate of adoption based on the type of adopters (“early” “early majority” “late majority”, and “laggards”), although this is recommended for future research. We focused on early adopters and the essence of their experiences from FND a priori, which is an approach that suspends the adoption of any theoretical structure or belief system prior to data collection and analysis [38].

### 2.3. Methods

This intrinsic case was conducted over the course of three academic years, including the pilot study. An intrinsic case study design was appropriate for our purpose, as this method focuses on a situation or a program and produces knowledge that is non-deterministic in nature—knowledge where “reality is constructed by individuals interacting with their social worlds” [39] (p.6). As is common in case study research, we drew our data from multiple sources and utilized purposeful sampling in order to illuminate our understanding of the FND experience [40]. The study was driven by the qualitative method as we were primarily interested in customers’, volunteers’, and leads’ subjective perceptions of their experience with FND. Measuring sustainability and food literacy was also important to us to build the case for FND; thus, we utilized a self-developed, validated, reliable survey for the pilot study. This research was approved by the UCCS Institutional Review Board (IRB) under the following numbers (IRB# 15-080 and 16-086).

### 2.4. Data Collection

#### 2.4.1. Participants

Purposeful, criterion-based, convenience samples of participants from four sources were selected for this study. Participants for the pilot study were selected from a pool of students, faculty, and staff from our university’s Nursing and Health Sciences departments; customers were selected if they purchased five FND meals or more (recruitment was open to students, faculty, and staff), and student volunteers were selected if they attended at least three FND meal preparation sessions to earn academic credit. Leads were sports nutrition graduate students funded to run FND. Purposeful sampling was deemed appropriate for this study because our aim was to engage participants who had experience with FND and could provide rich descriptions of their experiences [41].

Data were collected from academic years 2015/2016 to 2017/2018. Primary data sources were a survey, individual interviews, field notes, observations, materials, documents, and reports. It is recommended that multiple data sources be used to create the context surrounding participants’ lived experiences [40]. Our research team consisted of the lead researcher (lead author), a qualitative expert, and three graduate students. The team met frequently to debrief, initially identifying our biases and continually bracketing them throughout the course of the study, as paying attention to and controlling prior knowledge of a studied phenomenon is imperative toward the understanding of it [40].

#### 2.4.2. Pilot Study Year One

The pilot included a questionnaire with a validated food literacy survey administered at baseline before FND, which included an initial meals-to-go program in a small café setting before it moved to the center of campus in retail and residential dining.

The food literacy survey included questions related to planning, managing, selecting, preparing, and eating topics according to Vidgen and Gallegos [16] and topics pertaining to sustainability constructs such as equity, economics, and the environment as well as specific questions related to seasonality, geography, agriculture, and food crops. Vidgen and Gallegos defined food literacy based on the following: “Food literacy is the scaffolding that empowers individuals, households, communities, or nations to protect diet quality through change and strengthen dietary resilience over time. It is composed of a collection of inter-related knowledge, skills and behaviors required to plan, manage, select, prepare, and eat food to meet needs and determine intake” [16] (p. 54). Measuring food literacy remains a great challenge, since there are many definitions and constructs [17,18,19]. For the development of our survey, we were concerned that food literacy was addressed beyond the general understanding of food and nutrient interactions and their health benefits (nutrition literacy) and would include food systems and sustainability with the intention to integrate knowledge in food quality, seasonality, agricultural methods, environmental impact of food choices, and some aspects of food culture, behaviors, and skills, and perceptions of healthy and sustainable foods and eating (see Appendix A, for Survey).

The food literacy survey included 33 true/false/don’t know and multiple-choice items, with a maximum correct score of 33. The survey was subjected to content and criterion validity comprised of experts and a farm and food literate sample, respectively, and test/retest reliability according to the Flying Carrot Market Study, which was presented at the 2016 Sports Cardiovascular and Wellness (SCAN) Symposium [42]. For the food literacy survey, a score equaling or exceeding 75% was deemed food literate.

For the current pilot study, we used a convenience sample. Additional data collection during the pilot study included other demographic questions, questions pertaining to more sustainably and locally sourced food, participation in taste education, purchasing behavior, and willingness to pay more for local food. For the pilot study, we also quantified semester sales at a nearby campus café, as this was critical to bridge the gap between academic program goals and the fiscal responsibilities of self-operated dining services. Piloting whether a program makes economic sense is also important for DOI, as dining services are commonly categorized as revenue-generating services on college campuses and are more likely to adopt new ideas if they do not lose money. The purpose of the pilot study was to build the case for FND. In this sense, the pilot study was a needs assessment.

#### 2.4.3. Data Sources Year Two and Three

Year two and three included three different cohorts of participants with the main goal of exploring the overall experience with FND. We first focused on consistently (more than five meals) returning costumers (n = 10) in year two and on consistently returning (more than three sessions) undergraduate nutrition student volunteers (n = 8) and graduate student leads (n = 3) in year three. Graduate student leads were graduate students in sport nutrition and graduate assistants with SWELL who were paid through DHS. Participants in these three cohorts were interviewed based on the questions outlined in Table 1 (See Appendix B, Table A1). All interviews were conducted in person at the UCCS Farmhouse. Materials and documents produced for the time period spanning fall 2015–spring 2018 were reviewed. Together, the interviews and documents provided information on the program to help create a holistic understanding of participant engagement [39,43].

The three graduate student authors on this paper collected the pilot data and data from customers and student volunteers, while the lead researcher interviewed the leads. Interviews were audio-recorded and transcribed. Care was taken to make these inquires nonthreatening and easy to understand to allow participants to freely share their thoughts and opinions. Interviews were intentionally short and direct, although clarification and elaboration of meaning were sought when needed.

#### 2.4.4. Data Analysis and Statistics

Survey data were analyzed by counting each correct answer and deriving a total score. As there were no statistical differences among participant groups, except for sex, data were pooled and expressed descriptively overall and by sex (means ± standard deviation) and in frequencies of number/percent of scores above 75%. To evaluate the impacts of (1) taste education, (2) purchasing behavior, and (3) willingness to pay more for local food, food literacy scores were compared using independent *t*-tests (*p* < 0.05).

Interviews were analyzed by first identifying significant statements and quotes, followed by generating clusters of meaning from the statements to derive major themes and subthemes [41] related to the common experience of FND for the three cohorts. Word clouds were generated based on themes and subthemes (see Appendix C). We chose to use word clouds because they can be an effective tool for text analysis [44]. Themes weighted higher than subthemes (as they appeared more frequently during the analysis) and are highlighted in a larger font and different color. This representation has been shown most effective to draw the user’s attention to the words that had the most potency in terms of the meaning of the experience [45]. The word clouds we created provide a visual representation of the themes and sub-themes by participant group and are shown in Figure 3. They are placed along the continuum of food literacy and citizenship.

We also collected associated materials and documents generated during the course of the study and analyzed their content (e.g., recipe folders for each year (3), FND awards (3), SWELL annual reports (3), national presentations (3), internal white papers (3), and published articles referring to FND (2)). This analysis illustrated how food literacy was generated and disseminated. Then, interview and material and document data were triangulated. Distinct from triangulation as a form of mixed-method research (methodological triangulation), the process we used, called data triangulation, helped us identify different but complementary data on the same topic from the multiple sources of data we collected [40,41].

Finally, we engaged in the inductive process that Stake calls “propositional” or “naturalistic” generalization [43] (p. 86). We sought relationships among the themes that emerged from our three sources of data to build a summary or generalization of the FND experience, which builds food literacy first before individuals begin to transform into food citizens (see Figure 3), to which we refer as pathways. Utilizing professional expertise to interpret data and propose concepts or models for further study is a strength of QLR research [41].

Since the credibility and trustworthiness of data are critical to demonstrate rigorous QLR research, strategies recommended by Guba and Lincoln [46] and Creswell [41] were employed. The research team engaged in peer debriefing to achieve inter-rater reliability whereby each member thoroughly read the entire dataset and reached agreement on coding, themes, and key concepts that emerged from analysis. Disconfirming evidence, information that presented a perspective contrary to the norm was highlighted and discussed by the group as well.

## 3. Results

### 3.1. Pilot Study Results 

The pilot study was conducted in fall semester 2015 of the 2015/2016 academic year before FND was launched in fall 2016. A total of N = 75 (age 30 ± 13.2 yrs) participated in the pilot study, which included mostly females (n = 66), with a smaller sample of males (n = 9). The majority of participants (n = 58) were students, with the rest being faculty and staff (n = 17) in nursing and health sciences departments, including nutrition students. Sixty-four participants completed the food literacy survey. Scores by sex and university role are shown in Table 1.

Food literacy scores were lower in males (18.6 + 3.6) than females (22.1 + 4.2; *p* < 0.05). Food literacy scores were not related to participation in taste education (visited taste education: 22.2 ± 4.7 vs. 20.9 ± 3.9) but those who participated in taste education were more likely to purchase SWELL meals-to-go (*p* < 0.001). When evaluating purchasing behavior and food literacy, those who were purchasing SWELL meals-to-go had higher scores (23.8 ± 3.9) compared to those not purchasing meals (20.3 ± 4.0; *p* < 0.01). Food literacy scores were also higher in those willing to pay more for more local and sustainable food (23.2 ± 4.1) compared to those not willing to pay more (19.5 ± 4.8; *p* < 0.05). 

When evaluating food literacy scores, 75% of participants (n = 48) scored below a score of 75%. The range from lowest to highest score was 8–31 (24–94%). When asked about the interest in more locally grown and sustainable food served on campus, 64% responded positively, while 8% responded no, with the rest responding with “don’t know”. Over the course of the pilot study (Fall, 2015), there was an 18% increase in sales presumably from the increase in sales of SWELL meals-to-go.

### 3.2. Qualitative Findings from Customers, Volunteers, and Leads

To explicate meaning, case study researchers are seen as interpreters, which requires them to report their rendition or construction of reality related to the data gathered [40,41]. As such, this section tells the story of the key findings in narrative form from the authors’ perspectives. Our intent is to provide content from which further study and interpretation can grow. The three data sources interviewed are provided as thematic analysis below. These narratives also include the triangulation of the interview data from each data source with the data from the materials and documents selected for review. Verbatim quotes are woven into the narrative, which are identified using “*quotation marks*”. At the conclusion of this section, word clouds were developed to help summarize the experience for each of the three data sources. We conclude with a description of the essence of FND’s experience, leading to pathways from food literacy to citizenship (Figure 3).

#### 3.2.1. Thematic Analysis of Customers’ Experiences 

For year two, ten returning customers were interviewed regarding their experiences when visiting FND. Three major themes emerged: (1) fresh, flavorful food; (2) smiling, supportive staff; and (3) personal transformation. Table 2 shows the themes and subthemes in these three areas. The full table enriched with quotes from participants can be found in Appendix C (Table A2).

Theme 1 reflected customers’ comments about fresh, flavorful food, which pointed to the look and feel of FND. It was not difficult to spot FND in retail and residential dining, as the red-white checkered decor had a “*farmers market look*”. Customers commented that the station with local produce displayed and recipes to take was “*beautifully set up*” and looked “*just so fresh*”. The FND station also included announcements of Farmhouse Fridays (cooking) and other events and the menu itself. The menu changed weekly, with no rotation, and featured mostly regionally procured food except for olive oil and some spices. Customers were simply amazed by the variety and creativity, expressing this by saying “*this is delicious*”, *“prepared in a way that brings out different flavors”*, and *“(I) appreciate you do something different every week”*, even though some also acknowledged the *“taste difference”*. One customer mentioned that FND was *“the one healthy place on campus I can eat”*. FND meals were priced lower than other less healthful foods in the retail space, as an incentive; thus, customers mentioned the *“great price”* for such high-quality, locally sourced, healthy food.

Theme 2 focused on the smiling and supportive staff. One customer mentioned how *“helpful and knowledgeable and friendly”* the students were, that *“they always had a huge smile on their face”* and were visibly excited to being there, serving and supporting customers in eating more healthfully.

In retail, FND was serving two meals or a total of 150 servings per week to customers. Eating at FND seemed to help people eat better at home, as expressed by the following quote: *“I always feel really successful in my personal eating after I have finished my FND lunch”.* After students designed the menu, procured and harvested the food, prepared it, and served it, the verbal delivery was not only filled with information on sourcing, flavor, nutrition, and sustainability but also rich in passion and joy to serve food on campus that was truly from “next door” with the farmer’s story attached to it. One customer mentioned *“I like the way you describe the content of each meal”* and another said, *“You really get the impression that everyone helping out is really excited to be there and to serve food”*. 

Theme 3 highlighted personal transformation. FND led to personal transformation in shopping and cooking practices, partially because of exploring unfamiliar ingredients, as one customer put it *“expanding my horizons”*, shopping with more consciousness, and trialing new things *“with ideas for doing stuff at home”*. Most importantly, SWELL meals were delivered with education, which was highlighted by a customer saying, *“Everybody needs to know what healthy eating is all about”* and another adding, *“I didn’t know [healthy eating] could be as good as this is”*.

#### 3.2.2. Thematic Analysis of Volunteers’ Experiences 

For year three, eight volunteers were also interviewed regarding their experiences when working in FND. Three major themes emerged: (1) awareness; (2) knowledge and skills; and (3) connection. Table 3 shows the themes and subthemes in these three areas. The full table enriched with quotes from participants can be found in Appendix C (Table A3).

Theme 1 highlighted awareness to foods volunteers had not seen before. One volunteer said, “There were many vegetables that I didn’t know the names of, had never tried, [and] didn’t see in the grocery store before, and now I do.” Volunteers at FND were exposed to a variety of fresh, local, and seasonal ingredients and were taught by FND leads how to be creative in the kitchen and use these “new” foods to prepare delicious, healthy meals. Volunteers also got one meal at FND free of charge per week in appreciation for their work, which offered them an opportunity to see and live the experience full circle, witnessing their hard labor brought to fruition during service. Many of the recipes prepared by FND were plant-based, and this emphasis on plant-based eating influenced the volunteers’ own food choices, shifting some toward reducing their meat intake and incorporating more plants into their diets. One volunteer commented that FND “pushed me more toward a plant-based diet. I am Cuban so there’s meat in every single meal every day, and I don’t do that anymore. FND has definitely changed my habits and thoughts of what healthy eating is”.

Theme 2 focused on knowledge and skills pertaining to food and cooking. Volunteers articulated that they gained invaluable skills in the kitchen. One volunteer mentioned *“many students had never worked in foodservice before and just being in the kitchen, being around equipment was huge”*. They were exposed to a variety of local grains, beans, and produce, and when they volunteered, the FND leads would make sure the volunteers knew how to hold a knife properly as well as chop the produce, cook different grains, soak and cook dry beans, and navigate large commercial kitchens. Volunteers expressed gaining a deeper understanding of where the foods they prepared came from, for example saying, *“I did not know we grew a lot of stuff here at our farmhouse”* and *“I learned a lot about different foods and from where these foods were sourced”*. For example, the statements *“just being able to see where it comes from”* and *“knowing that everything is locally sourced”* related to FND’s procurement commitment, which was always a critical topic, along with food safety skills, when the menu preparation with volunteers started. Furthermore, volunteering with FND gave individuals exposure to and the skills necessary to educate on topics of local food and cooking, such that this newly acquired knowledge could transfer into their future professions. This was summarized by the following: *“FND added another aspect of nutrition to my knowledge and to my practical use when I become a registered dietitian”* and *“It’s something that I can utilize, even with athletes that I know could benefit from it”.*

Theme 3 is related to personal and inter-personal connections. Food preparation was at the same time each week when students would come together to work toward a common goal: preparing delicious local food for the campus community. One volunteer said, *“I love this program… FND creates community. It brings us all together”*. FND not only connected individuals, but it also helped them connect with food and build a better relationship with food. *“Through FND, I learned more about food and got a better relationship with food.” “It was good to see a different side of foodservice that’s more appreciative and more exciting to work with.”* As a result of the freshly picked produce, the kitchen smelled not of grease or sugary bakery items but of irresistible flavors and food that looked like it came straight out of the soil, vibrant in color and life, rather than pale and out of a plastic bag. This surprisingly new connection with food also led individuals to become more mindful. *“Healthy eating is more than just eating a fruit and vegetable with each meal. I think it has more to do with where it comes from, how it was grown, and the people involved in that process.”* Synthesizing the experience of FND for volunteers was probably best expressed by the chosen title for the graduate project reporting on the experience of volunteers with the statement *“the future dietitian cooks”*!

#### 3.2.3. Thematic Analysis of Leads’ Experiences

For year three, three leads were interviewed regarding their experiences when working in FND. Three major themes emerged: (1) becoming a leader; (2) peer teaching and learning; and (3) food systems knowledge, skills, and advocacy. Table 4 shows the themes and subthemes in these three areas. The full table enriched with quotes from participants can be found in Appendix C (Table A4).

Theme 1 was focused on becoming a leader, with leaders having to hold a lot of moving parts including planning and procurement, harvesting, cooking food, serving meals, marketing the menu, and educational messaging. One lead expressed it by saying, “*As a leader, you just have to always put that face on and make decisions. People are counting on you. At the beginning, it was overwhelming, juggling so many roles in the kitchen. Sometimes, I just wanted to be in a corner with a cutting board and cut the beets*”. Handling large quantities of farm-fresh food, harvested just hours before preparation, teaching concepts and skills such as food safety and knife skills to volunteers while also delegating tasks, dealing with challenges such as seasonal supply or cooking with unknown ingredients, and creatively adjusting the recipe based on familiar customer expectation was a big responsibility that required full effort and commitment. Leads expressed needing to consistently *“stay(ing) engaged, focused, and fully committed to the work*”. There were so many challenges to manage. “*There was so much food in the fall to prepare, it would take three hours just to wash the food and we had not even started cooking it", “cooking with unknown ingredients”* or *“odd balls like the tomatillo and how do we make that part of the meal*”. In commercial kitchens on this scale, it is usually the menu and recipe that guides the cooking. In FND, this was somewhat flipped, where the ingredients, especially if unexpected or unknown, lead the menu and required recipe adjustments. While this was a challenge, it also offered opportunities for creativity. FND leads were expressing their excitement even if they were under pressure. *“To be put in that leadership and management role as a graduate assistant was a lot of pressure, but it also made me rise to the occasion. And you do things that you did not think you could do. The commitment, I think, was really amazing. And you get out of it what you put into it and I cared a whole lot about it. I would stay awake at night and think about it. I would draw diagrams of how I was going to set up the bay at FND the next day. I would stay up at night and do that. I loved it.”*

Theme 2, peer teaching and learning, was especially informed by the lead students’ steep learning curve, feeling inspired, and yearning to pass it on to others. While in the kitchen with volunteers, FND leads exercised the highest order of command as leaders and managers, following their action plans and ensuring safe practices, while also teaching what they had just learned to their student peers. *“Running a foodservice operation with orders, money, procurement, catering … brought about necessary collaboration*” as one lead expressed in preparation for the weekly action plan, which required *“a lot of thinking”*, *“including our own ideas”* not just *“flipping cookbooks or pulling recipes”* off the internet. *“It had an impact on me, so I like(d) shar(ing) that.”*

The kitchen work was not comparable with handling culinary work in a conventional supply chain. There were “*bins and boxes with food straight from the earth”*. This new learning was often overwhelmingly challenging. Yet, FND’s cyclic experience enabled a continuous, reciprocal dialogue from teacher to learner about farming and gardening, opportunities for creativity through all the senses, and a platform for students to *“learn from mistakes”* without significant consequences due to the academic environment. All leads expressed similarly how *“working on the farm”* and *“farming and gardening”* meant they were *“able to teach skills in the kitchen”*.

Theme 3, food systems knowledge, skills, and advocacy, emphasized the enhanced understanding and navigation within the food value chain. First, common among all leads was that FND built food literacy *“opening eyes to food, cooking, and gardening”*, understanding *“seasonality”*, and becoming more aware that nutritiousness depends, at least to some degree, on *“trying to learn where food comes from”*—the food’s origin and when and how it was grown. Comments such as *“I never tied local food together with nutrition”* reflect significant curricular gaps in nutrition and dietetics. Secondarily, building relationships and shifting values through the newly developed relationships with farmers were also important. Leads identified new relational values of food, associating food with *“planetary health”* and expanding definitions of healthy eating from calories and nutrients, such as protein, to food, the food system, and sustainability. Participating on farms, experiencing the sweat from harvesting and what goes into food production, both in urban gardens and on rural farms, provided a deeper level of understanding of the food system and food value chain, in one lead’s words, *“why it is important that we use these foods”*, in the support of the regional community, food security, and conservation of open lands for agriculture.

To summarize the findings from year three, FND leads learned to align their thinking, planning, and procurement with the constraints of food availability in a regional context. What used to be common sense in seasonality had to be relearned. The interaction with local farmers and soiling their own hands weekly while physically harvesting what was fresh taught them so many aspects about food that started with soil, water, climate, biodiversity, and seeds. Environmental challenges, such as drought in the West or elevation and the length of the growing season were topics that brought more understanding to the limited availability or the type of food, its flavor, culinary use, and nutritional benefits along with its origin—sometimes being indigenous or ancient. For FND volunteers, food preparation led by their graduate peers increased their awareness to the foods grown in Colorado and the climate-friendly, healthy SWELL meals and introduced some level of environmental awareness and food production while increasing practical knowledge and skills in institutional foodservice settings. FND also brought them a renewed sense of community and food connections. They too learned about the food system they were working in, its interdependence, and collaborative spirit while working together in the kitchen, of course, distracted by smells, sights, and flavors from this fresh food. What was tough for the leaders to juggle seemed to be softened by the joy of cooking together, which was so well reflected by the consistent return of volunteers.

The quote below sums up the experience by one of the most experienced student leads, indicating a deeper understanding through immersion in FND.


*“Being able to have my hands in the garden, there was something very special that I was able to transfer into the kitchen. Coming with bins and bins and bins of fresh produce from the garden, much of them I never had true exposure to. You hear about leeks, tomatillos, many kinds of peppers. But to be able to harvest, then take it in an electric cart over to campus and learn how to cook with it, which part of it you eat and which part you don’t eat. Can you eat the seeds? Do you take them out? Are they spicy? Sweet? What about the skin? How do you cook it? And it was so fresh. I had never dealt with such fresh produce before. Never in my life had I taken all the food from the garden and then cooked it. That was brand new. Being able to have experience with food and also learning about the local farms and identify(ing) which farms have what food and at what time. Learning the seasonal calendar was a learning curve for me. But the food aspect of it is amazing. Learning to cook with different herbs, how the fresh food can really make a dish. You don’t have to do a lot to it when the produce is so fresh and flavorful. You don’t need to add a ton of spices or salt and you don’t need to cook it in any special way to make it taste great. And I think that was something that really was amazing that both the customers and I were able to recognize quickly. The customers would express how flavorful and amazing the food was, and we would tell them, it’s just five ingredients all tossed together. A light bulb went off for them and for me. I started to know food differently, despite my previous culinary and sustainability training. It just goes to show that learning it in the classroom is one thing, and hearing about it is one thing, but actually being fully submerged in doing it is another thing.”*


The peer teaching and learning from leads to volunteers touched on all components of food literacy, including critical food literacy. The recipe development process followed by the ambitious action plan from harvest to service required trans-disciplinary thinking and action. The menu included both environmental and health objectives, but it was additionally confined by the availability of products and taught the students lessons of restraint, flexibility, creativity, and ignited forgotten skills in home economics to extend the seasons of food (e.g., canning, freezing). It also involved the transport of harvested food for “mise en place” in commercial kitchens and unpacking, combining, and bulking up recipes, costing the menu based on purchases, while keeping track of farmer names and locations. The educational portion often asked for interaction with the farmer or food hub, learning about their stories, the history and culture of their crops, experiencing a small piece of the lives of American family farmers, and their hardship of staying afloat. Society’s romanticism of farming or gardening was replaced by sober knowledge, empathy, and arising commitment to participate in and contribute to a thriving food system. As one lead put it *“how much work goes into food production is a humbling experience. I have more respect for food and want to do it justice”.*

#### 3.2.4. Pathways from Food Literacy to Citizenship

The themes from our data sources showed a progression (Figure 3) from food literacy to citizenship, which we term pathways for the purpose of in this study. The pathways reflect FND’s ability to (1) help customers re-imagine what healthy food is all about, (2) create and gain greater and new food awareness, (3) build knowledge and skills from farm to kitchen through experiential learning in value chains, and (4) reconnect to food, planet/nature, and each other. From customers to volunteers and leads, food literacy and aspects of food citizenship were present. FND promoted food literacy in customers, while in volunteers and leads, food literacy was a latent effect of FND experiences. However, considering food citizenship, the leads’ themes illustrated broader understanding of the food system in general, and value versus supply chains specifically, as they relate to the environment and social causes. The responsibility of bringing SWELL meals to campus eaters within the parameters and expectations of DHS invoked pressure to make a value chain work in a mainstream institutional dining environment. To receive leadership roles in graduate school is one aspect of student learning but to be committed to all the extra steps involved, as shown in FND (see Figure 1 and Figure 2) requires re-imagining food and being inspired by its many values, especially those external to oneself.

Figure 3 shows word clouds for themes and subthemes of customers, volunteers, and leads summarizing the individual and combined findings visually.

## 4. Discussion 

This intrinsic case study described experiences in customers eating at and students volunteering and leading FND, a local food establishment on a college campus. FND involved a cyclic process from farm to kitchen and table with a fully student-led program, producing environmentally responsible meals, balancing carefully flavor and nutrition in menus and recipes, while discovering the foods of Colorado, the seasonal and variable supply of produce, the cross walk from campus farm to kitchen, and the infinite dialogue among student peers—teaching, learning from challenges, and rising to the opportunities of bringing this food to campus eaters. From customers to volunteers and leads, themes we identified highlight experiences from FND as life-changing.

In addition to a vigorous science and didactic curriculum, nutrition and dietetics programs cover many, if not all, aspects of nutrition and food literacy, but the field is especially focused on functional and interactive components. For example, nutrition students research and learn the scientific and inter-personal dimensions of nutrition in health and disease and how to work with patients to improve eating habits and promote health or treat disease using medical nutrition therapy and the nutrition care process [47]. Critical food literacy offers a deeper discourse with reflection and consideration of broader issues in a social or environmental context [18]. Critical components of food literacy in the dietetics profession are especially strong related to advocacy and leadership work of the association and its professionals, with many opportunities for young dietitians to get involved. However, the inclusion of sustainability related to environmental, social, and economic aspects of food systems work from research to practice, advocacy, and leadership has only recently started [6,8,9,10,11], with many opportunities for curricular inclusion yet to be explored. This intrinsic case study, capturing experiences with FND, involved mainly nutrition students immersed in food value chains [26] in which students acquired broad aspects of food literacy and citizenship, complementing their nutrition curriculum.

There is a recognized progression from food literacy to food citizenship. Critical food literacy is conceptualized as a kind of food citizenship where work with food extends beyond oneself and expands to societal dynamics [18,48,49]. Most recently, Rowat and colleagues highlighted critical food literacy using a multi-dimensional approach and leveraging academic settings and their formative years to train the next generation of food citizens [23,27]. FND is one of those training grounds for college students, but it was not only the leads and volunteers (nutrition students) whose themes identified personal transformation and some level of critical food literacy. Even the customers who ate at FND were impacted. They commented on its freshness and taste, the welcoming and inspiring setting, and the knowledge delivered with each plate.

Food literacy is desperately needed in university students, including those students preparing for health professions [50]. From our food literacy survey, more than 75% of the participants in our pilot study, who were mainly nursing and health sciences students, scored below 75%. Females scored higher than males, which should not be surprising, as a recent scoping review shows that being female is a predictor of healthy eating on college campuses [51]. The authors also highlighted influential factors for healthy eating which were food literacy-related constructs such as nutrition knowledge but also perceived happiness, stress, and students valuing healthy eating as important. They also found that students’ openness to new experiences [51] can influence healthy eating. FND is one of those new experiences for students on college campuses with food literacy opportunities arising from dining services. This was recently identified in a qualitative study in which undergraduate students reported developing and applying food literacy from eating in the dining halls, especially where helpful messaging and healthful menus were included [27]. That customers of FND mentioned the friendly and knowledgeable staff highlights the value of creating a non-intimidating food literacy environment staffed by “nutrition experts in training” who were peers. Peer teaching is known as an effective strategy in young people, especially considering generic skills [52] such as healthy eating.

While FND required campus resources and full commitment of their leads with many challenges to overcome, such establishments can live on college campuses and make dining experiences unique. Our pilot data were needed to launch FND, as we were able to show the need for food literacy in students, faculty, and staff using our survey data. Those purchasing meals-to-go were interested in and willing to pay more for healthier and more sustainable food options, which demonstrated higher food literacy scores. We were also able to show an almost 20% increase in sales with meals-to-go, all of which persuaded DHS to start FND. Unfortunately, socio-economic status is often predictive of healthy eating, also in college students [50], and food insecurity on college campuses is high [53]. It has to be stated that FND-like dining operations do not have to generate revenue, as their value proposition is expansive and goes beyond simply getting a healthy meal on campus. In addition, small pricing adjustments such as making unhealthy choices slightly more expensive can help meet budgets in institutions. As long as these operations cover most expenses, they are win–win–win solutions for both academic programs and dining services and they encourage the institution’s participation in the regional food system.

FND experiences taught volunteers and leads, who were nutrition students, new strategies for healthy eating from sustainable food systems according to what we believed was meant, at least in part, with a call to action by the EAT-Lancet Commission on Food, Planet, and Health [4]. Developing menus and educational messaging, while also proving that this kind of cooking and eating is doable, delicious, and adoptable not only served as encouragement for customers but also instilled a level of self-efficacy and confidence that solutions on the consumer side are within reach, at least for early adopters. While we are not aware of similar studies targeting nutrition students in university settings, others have shown that acquiring food systems knowledge from academic courses [30,31] can positively impact behavior change pertaining to both healthy eating and sustainability in young people. Moreover, one of the most critical issues in addressing climate change through the lens of the consumer is their willingness to eat more plant-based meals with less or no meat [4,54]. Our data show that if plant-based and plant-forward menus can be modeled, such as in FND, and consumers (including FNDs’ leads and volunteers) acquire the knowledge and skills, obtain the tools (e.g., recipe ideas), and gain access, changes in eating habits are possible. Thus, experiences described from FND are promising and in line with others [27,28,29,30,31,32,33].

Experiential learning opportunities are well-recognized pedagogical strategies to deepen students’ understanding of classroom materials [15,55,56]. In this case, FND was most aligned with foodservice management, food systems, and food culture courses, but the cyclic experience of a food value chain offers unexpected tools for healthy eating from sustainable food systems and opportunities to shift values. In rethinking the values of nature, Chan et al. [24] illustrate the experiences needed for young people in order for relational values of nature to be rediscovered. Our data on FND’s volunteers and leads showed that working in value-based food systems can offer such opportunities, especially for nutrition students, to learn more about the environmental and social issues of the food chain and shift their values. FND’s charge to educate campus eaters on healthy and sustainable food connections brought attention to the environmental impact of agriculture and the urgency to address climate change and protect nature for young people [57].

Agriculture has a significant impact on climate change. Food production and consumption are responsible for 26% of all greenhouse gas emissions [54], decreased marine and terrestrial biodiversity [58], soil nutrient losses, and water shortages [54,58]. Especially since COVID19, many food systems issues, such as industrial food production methods, have been exposed as hotspots, being extractive and damaging [25]. Combined with population growth and climate change impacts, there is no other sector as heavily burdened as the food system with a risk for unprecedented future food shortages and widening food insecurity, especially in socially disadvantaged segments of the population [4]. In addition, health and disease are both related to climate change [59] and have recently been highlighted as syndemic [60], which means they are to be addressed from multiple sectors with skillful collaborative approaches. While only a small example of multi-disciplinarity, our data on FND show how the integration of experiential learning in campus dining operations leveraging food value chains can meaningfully enhance nutrition students’ understanding of food, health, and sustainability connections and fill existing gaps in the curriculum for RDs. Such multi-disciplinary experiential learning opportunities in nutrition curricula have also been identified as urgently needed by others [61].

Especially the FND leads seemed to have gained a deeper understanding of the current food system and learned how to engage in and help build an alternative value chain alongside the institution’s mainstream suppliers. Based on interview questions such as “How did FND change the way you think about and act around food?” and “What was most important to you about FND?”, leads expressed the pressure and challenges of FND but also the profound learning opportunities being fully submerged in the food value chain. Their daily interaction with DHS brought many challenges and opportunities for student leads to apply critical thinking, problem solving, and decision making. For example, students had to solve logistical issues, learn about procurement policies, collaborate, and effectively communicate with each other and DHS. They also saw the hard and often less fulfilling work of foodservice personnel and began to understand the uneven distribution of the food dollar along the supply chain all the way to the farmer. They recognized the current disconnect of foodservice workers from the food source. Discerning differences in supply vs. value chains [26], recognizing strengths and weakness of both with respect to social, economic, and environmental sustainability [62], students in FND learned to speak out on social and political dimensions of food, as described by Rowat et al. [23]. Whereas FND required full effort and commitment from its leaders, it also introduced complex food systems issues, not to be solved by FND, but with hopeful preparation for future work.

Change agents are now needed to address food systems complexity. However, health and nutrition professionals will not simply be able to opt into specialized practice in sustainable food systems [9]. These professionals, along with other health professionals [50], will have to be prepared to act from within to support safer, more accessible, and fresher more transformative food systems, especially if addressing rural health disparities, food insecurity, and changes in eating behaviors. FND’s work connected nutrition students to farms and food through their work from within the regional food system of rural Colorado. They learned to relate to food intra and inter-personally as well as ethically. They expanded their food values and shifted how they thought and acted around food and farming, cooking, and eating, with greater relational as opposed to simply instrumental terms [24]. Thus, our study shows transformation in students through experiential learning in sustainable, healthy food systems and provides a good example of what curricular integration in nutrition can look like while leveraging dining services as living–learning labs.


*“One must begin in one’s own life the private solutions that can only in turn become public solutions.”*
Wendell Berry [63]

## 5. Limitations

This three-year qualitative analysis of FND has limitations. First, our sample sizes for the pilot study and the interviews were small. The main researcher was the faculty advisor of FND and directed UCCS’ food journeys since 2009 [35]. FND had six student lead pairs (12 leads in total), who were all funded by DHS. This study only captured a two-year block. The results are informed through the interviews, materials, and documents from these two years, but they are also enriched by observations of an additional four-year block that helped the lead researcher understand the learning environment of FND and the progression in food literacy. Overall, over 50 nutrition student volunteers have been involved in FND, impacting several hundreds of undergraduate students, including freshmen in both retail and residential dining. Since 2016, graduates of FND have returned to UCCS for Grain School. While not included in the interviews, the voices of these alumni, or grain citizens, have been critical in this broader analysis from food literacy to citizenship.

Another limitation of this study is the food literacy survey. This survey was developed for another project and not FND and college students specifically. Nevertheless, the survey we developed captured what we needed to build the case for FND.

## 6. Recommendation

That health and nutrition professionals are part of the future food systems workforce is in the hands of those writing curriculum and competencies and in innovations for post-graduate training that are multi-disciplinary in nature and use systems thinking, as illustrated recently by Ingram and colleagues [14]. On the most rudimentary level, curricular inclusions in health and biomedical sciences should integrate concepts of sustainability [64,65] and align with the UN Sustainable Development Goals [66] and the Planetary Boundary Frameworks [67]. However, nutrition sciences and public health nutrition must embrace sustainability [68] and expand and learn from agriculturalists, among many other disciplines [14]. Recently, Vettori and colleagues included “closer connection to producers and agricultural and farming towns” as an antecedent to food literacy [19]. While food systems solutions are often expected to originate from technological and agricultural innovations, there is great urgency to think and act more inclusively and look at the food system at large, bringing new ways of learning [14] and indigenous ways of knowing [4], aligned with institutional goals and resources, and in collaboration with multi-disciplinary academic and community sectors [14], to equip food systems analysts with the tools to get to work.

While there is good momentum currently in expanding the sustainability competencies of dietitians in North America [5,6,7,8,9], Europe [10], and globally [11], to respond to the calls to transform the food system from production to consumption [3,4], these competencies must move to the center of the curriculum such that food systems thinking and sustainable development is integrated throughout and not sidelined. That FND experiences were some of the first food system trainings for volunteers and graduate students highlights the gap in curricular content. Some of the student leads said it themselves: *“changing RD curriculum to integrate agriculture”*. Our research, describing the experiences from FND, provides a start. Unique to QLR is that categories of meaning, here expressed as themes and subthemes, may be used in developing assessments or competencies for the purpose of developing nutrition curricula to include sustainable food systems experiences.

## 7. Future Research

Further research is needed on the long-term impacts of food systems courses and co-curricular activities such as FND in students and beyond the early adopters we studied. It is of interest to better understand what experiences lead to all aspects of food literacy and transformation to food citizenship, especially in nutrition students. Finally, more needs to be known about how these experiences in nutrition students in college transfer to food systems work in dietitians in the future.

## 8. Conclusions

This intrinsic case study focused on a student-run, farm-to-table dining operation, called FND, on a college campus and experiences of early adopters, including customers eating at and nutrition students volunteering for and leading the operation. Our pilot data set the stage for this FND as food literacy was low, but interest in and purchases of alternative food options high. Interview themes in customers related to fresh and flavorful food, smiling and supportive staff, and personal food transformation. Volunteer themes highlighted greater awareness of new foods and plant-based approaches, cooking skills, and connection. Lead themes focused on leadership, peer teaching and learning, and food systems knowledge and advocacy. Through this case study’s inductive process, pathways from food literacy to citizenship emerged. Experiencing and becoming part of the food value chain through FND built food literacy, shifted values, and transformed participants into food citizens.

## Figures and Tables

**Figure 1 ijerph-18-00534-f001:**
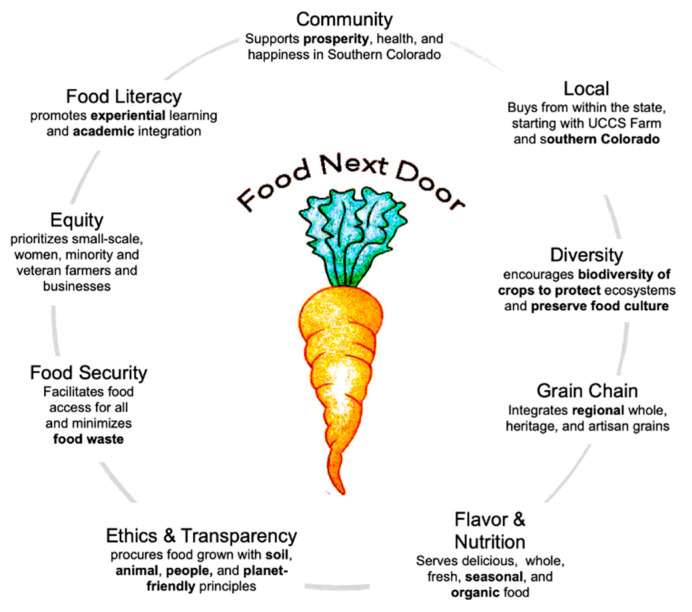
Food Next Door values for sustainable sourcing, adapted from Meyer, 2020 [35] with permission.

**Figure 2 ijerph-18-00534-f002:**
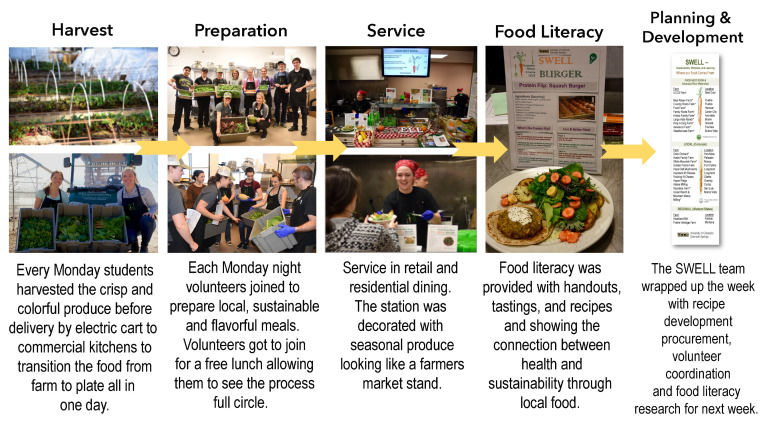
Weekly cyclic process managed by two Sustainability, Wellness & Learning (SWELL) graduate assistants (hired by Dining and Hospitality Services) who are in the Department of Human Physiology and Nutrition and specifically, in the Sport Nutrition Graduate Program.

**Figure 3 ijerph-18-00534-f003:**
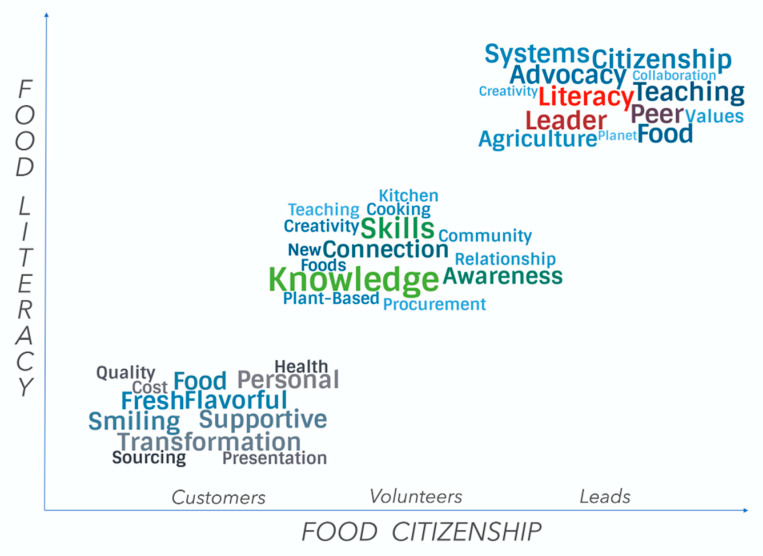
Themes and subthemes for customers, volunteers, and leads.

**Table 1 ijerph-18-00534-t001:** Food literacy scores by sex and campus roles.

Parameter	Total Sample	Food Literacy Score
	(n = 75)	(n = 64)
	n (%)	mean ± SD
Sex		
Male	9 (12)	18.6 ± 3.6
Female	66 (88)	22.1 ± 4.2 *
Main Role at University		
Undergraduate Student	43 (57)	20.7 ± 3.6
Graduate Student	15 (20)	24.8 ± 3.8
Faculty	15 (20)	21.5 ± 5.3
Staff	2 (3)	19.5 ± 4.3
Total Sample		21.7 ± 4.3

* significant at *p*-value < 0.05 (independent *t*-test).

**Table 2 ijerph-18-00534-t002:** Experiences described by Food Next Door (FND) customers (n = 10) in year two.

Themes	Sub-Themes
Fresh, Flavorful Food	Cost
	Flavor, Presentation, Creativity, Variety
	Food Quality, Health Benefits
	Sourcing
Smiling, Supportive Staff	Knowledgeable
	Passionate
	Friendly
Personal Transformation	Shopping Routines
	Cooking Experimentation
	Knowledge

**Table 3 ijerph-18-00534-t003:** Experiences described by Food Next Door (FND) undergraduate student volunteers (n = 8) in year three.

Themes	Sub-Themes
Awareness	New Foods
	Creativity
	Plant-Based
Knowledge and Skills	Cooking Skills
	Procurement
	Teaching Others
Connection	Creating Community
	Better Food Relationship
	Mindfulness

**Table 4 ijerph-18-00534-t004:** Experiences described by Food Next Door (FND) in graduate student leads (n = 3) in year three.

Themes	Subthemes
Becoming a Leader	Full Effort and Commitment
	Creativity
	Dealing with Challenges
Peer Teaching and Learning	Passing it on
	Learning New Skills
	Collaboration
Food Systems Knowledge, Skills, and Advocacy	Food Literacy
	Building Relationships, Shifting Values
	Food Citizenship

## Data Availability

The data presented in this study are available on request from the corresponding author.

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
