# Peer review of "Food Next Door: From Food Literacy to Citizenship on a College Campus"

_ijerph, 2021, doi:10.3390/ijerph18020534_

Round 1

Reviewer 1 Report

Dear authors and editors,

Thank you for the opportunity to review this interesting article.  This piece is an account of a small-scale, 3-year study on a University of Colorado, Colorado Springs campus program to promote healthy eating and local food.  The authors note that curricula in nutrition and dietetics often do not intersect with those in sustainable agriculture, and the “Food Next Door” (FND) program seeks to bridge that gap by focusing on “food literacy” and “food citizenship.”  FND aims to allow students to work with locally-grown food (from nearby farms in southern Colorado, and the UCCS campus farm) directly after harvest, and prepare it for retail to students.

The data consists of a pilot survey of 75 on food literacy, and 21 total interviews (over three years) coded for themes.  Interviews were conducted with customers of FND, undergraduate student volunteers, and graduate student program leads. While there is little ethnographic description of the FND, I feel that would strengthen this piece, given the small sample size.

Of course, it is inherently difficult with a short-term study to measure what impacts a program might have on eating patterns or understandings of food systems.  Nevertheless, the data presented are encouraging that campus eating programs can positively impact not only those who eat, but also those who cook and prepare that food.  It is also a compelling example of how campus dining systems might be integrated with nutrition curricula in a meaningful way.  This study embraces the complexity of teaching about food as a multi-disciplinary and experiential process.

In all, this study is a useful contribution to the growing literature on improving campus dining systems, as well as food systems pedagogy. I look forward to reading more research emerging from this program.

Minor issues:

Pg 1 line 41.  The “I” (Meyers?) should be made explicit, since there are multiple authors.

Line 658:  Again, the “main researcher” should be identified clearly by surname, as author-listing conventions are not the same in all fields.

Author Response

Thank you for your encouraging comments and remarks. We appreciate your evaluation. We made extensive changes to the manuscript based on Reviewer 2. We also wanted to thank you for the recommendation to add more information on FND specifically and ethnographically. You can now see this in lines starting at 347 but also above.

Minor issues:

Pg 1 line 41.  The “I” (Meyers?) should be made explicit, since there are multiple authors.

This was removed.

Line 658:  Again, the “main researcher” should be identified clearly by surname, as author-listing conventions are not the same in all fields.

We removed this based on the recommendation of reviewer 2. Thank you for your suggestion.

Reviewer 2 Report

Overall, I find this intrinsic case study interesting and relevant. There is a need to have empirical studies on re-connecting consumers to the food chain. However, the paper needs major revisions to improve it.  I summarize the key revisions as: (1) DOI needs to be linked to the research design; (2) results and the first half of the discussion should be re-organized and combined. Use the DOI to frame the key topics collected through multiple data sources and then organize results to follow it. The aim was to evaluate “pathways” of food literacy and citizenship and that has yet to be achieved in the paper; (3) the style of writing should be academic rather than narrative. It is written in a tone that would be appropriate for a book chapter but isn’t appropriate for an academic paper.

See more detail below:

Abstract has all the necessary components but is missing some transitional phrases that would better link the sentences together—and make it easier to read.

First two paragraphs are written in a narrative style—not an academic format. It is more appropriate for a book chapter than an academic journal.

The introduction could be improved in a few places including the final concluding paragraph. I’m not sure why “(and farm)” is inserted—seems tangential. The significance/relevance of the research topic should be more clearly articulated. This would help us better understand why this gap in research is important to address. 

“…the aim of this study was to evaluate pathways of food literacy and citizenship in university students, faculty, and staff experiencing eating at, volunteering for, and leading Food Next Door, a student-run, local food establishment, on a university campus.”

The research population isn’t clear: is it “students, faculty, staff” or “customers, volunteers, leads” (ln 219) And, what are "pathways"?

Methods overall could be improved to reduce redundancies and to better clarify the timeline of the study. For example, what was the purpose of the pilot study? Where did interviews occur? An image / images of the farm/cafe would be helpful. I find the syntax awkward, which makes it difficult to read. It’s unclear if pilot results were used to design Phases 2 & 3.  There isn’t a clear link between DOI and methods / research design / analysis. It’s confusing as to whether FND actively educates and promotes food literacy or if it is a latent effect? What are the research questions or objectives guiding research design? Where participants tracked to see if there was an increase in their food literacy / citizenship?  If not, how do we know which “pathways” impact food literacy / citizenship? Without that, it is difficult to interpret results.

Paragraph ln 331-339: Basically, this is reporting that those with higher food literacy scores were more likely to purchase and pay more for the food. I suspect it is more self-selection rather than causation…

3.2.1, results need to be more directly linked to the research aims. I find much of the summary tangential—and frankly not that interesting. For example, the fact that customers return because the staff is friendly is inconsequential.

3.2.2 & 3.2.3 are more aligned with research aims but could be more clearly organized

Much of the discussion belongs in findings—it is the summary of data analysis.  The discussion should focus on higher level and key findings rather than a more narrative and specific summary of results. It should also link key findings to the literature in general, which starts to happen in Ln 586 onwards.

Re: Limitations: a small sample is not necessarily a limitation, particularly with qualitative research. It would be better to give the audience an idea of how many people were involved with FND to get a sense of how the sample was representative. Re: the food literacy survey—how were reliability and validity ensured? What about construct validity? Finally, the fact that the author lives on a farm is not a bias—the study wasn’t about farms but about literacy. The two are not the same.

Recommendations: where are they? This section is vague. Suggestion to re-write with specific actionable recommendations for practice, research, and theory. Or delete.

At the end of the paper, I am left wondering what ARE the pathways to food literacy and citizenship?

Conclusion: The authors haven’t clearly provided evidence to support the statement Ln 708-709.  Conclusion should be improved.

In reading Appendix A, I suggest organizing findings around the research questions.  These should also clearly link to the DOI in the methods section.

In general, while the narrative style is accessible, it is not appropriate for an academic journal. It lessens the authors’ credibility by implying subtle biased opinion throughout.

Specific items:

  • Abstract: what does “intrinsic” mean, i.e. intrinsic case study? Why is this representative of an intrinsic case study as opposed to, say, a critical case study?
  • Numerals 1-10 should generally be written out. Exceptions are when it is with a larger number such as Ln 65-66
  • Ln 71-71: Citation?
  • Ln 75: (Wicked!) functions as both a reference to wicked problems as well as a value-laden term…not appropriate for an academic paper.
  • Ln 81: Citation for the assumption that professionals have high food literacy?
  • Ln 120-121 is distracting; omit
  • Ln 124-125: Avoid single sentence paragraphs.
  • Paragraph Ln 161-186 is really long; is all this background information relevant and needed for this paper?
  • Ln 195: Citations???
  • Why was purposive sampling used (besides to illuminate understanding)—what were inclusion/exclusion criteria?
  • 2.1 It says there were four sources for selecting participants but three are given
  • Ln 261: Not necessary to state that a conference presentation/paper remains unpublished
  • Paragraph Ln 263-270: Some of this should be in findings, for example, how many participants and dates of data collection
  • Table 1: Students above/below C—are data in the correct column?
  • Table 2: What is the relevance to the study? Increased sales isn’t indicative of increased literacy.
  • Ln 346-347: Not necessary to include the sentence beginning “Verbatim quotes”; omit
  • Ln 507: And compost???
  • Ln 515-534: This quote is unusually long and is presented as raw and unanalyzed data in the discussion—paraphrase and move to results.
  • Ln 600: What do the authors mean by “progression”? Perhaps “hierarchy” is a more appropriate term
  • Ln 601: How can the authors claim a “deeper understanding” occurred? Authors need to more explicitly describe interview questions so we know how change was measured.
  • Paragraph Ln 663-671: Suggestion to omit; not relevant

Author Response

Thank you very much for your constructive feedback. We have made substantial changes to the manuscript based on your review. In general, according to the numericals above, we have addressed 1) through 3). I will briefly summarize below and respond point-by-point using the reviewer’s requests.

  • We have made several adjustments to link the paper with DOI. However, our study was not a systematic DOI inquiry but rather exploratory or descriptive in nature regarding the experiences gained from Food Next Door (FND) in early adopters such as the customers eating at FND and volunteers and leads working for FND. We clarified this in the paper in the following sections:
  • We completely rearranged the results and discussion. We renamed qualitative results into findings and shifted the narratives into this section and tightened the discussion with focus on other research. We also included the ‘pathways’ into the purpose of the study and added a section in the findings that links the ‘pathways’ to figure 3, better explaining what we propose as ‘pathways’ based on our triangulation of the interview data and other data sources.
  • Because of this paper being a qualitative paper, we moved the narrative nature into the findings to reserve narration for the findings rather than elsewhere, especially the discussion. We also adopted most of the suggestions and changed the writing style to more academic writing.

See more detail below:

Abstract has all the necessary components but is missing some transitional phrases that would better link the sentences together—and make it easier to read.

Thank you for your comments. Due to the brevity of the abstract we were careful with word choices but have integrated your feedback. The abstract now reads like this:

Abstract: Industrial agriculture and food corporations have produced an abundance of food highly processed, nutritionally poor, and environmentally burdensome. As part of a healthy campus initiative, generated to address these and other food production and consumption dilemmas, a student-run “local & sustainable” food establishment called Food Next Door (FND) was created. This intrinsic case study evaluated food literacy in health science students, faculty, and staff first as pilot to build the case for FND and further explicated customers’, volunteers’, and leads’ experiences with FND, identifying potential pathways from food literacy to citizenship. Ten returning customers, eight recurring nutrition student volunteers, and three graduate student leads participated in interviews that were analyzed for themes and sub-themes. Findings show a progression in themes. Customers’ experiences highlight FND’s fresh, flavorful food, smiling and supportive staff, and personal transformation. Volunteers’ themes identified greater awareness of new foods and plant-based eating, acquiring new knowledge and skills in commercial kitchens, and deepening their connection to food, each other, and to where food comes from. Leads’ themes show opportunities to gain managerial skills, a deeper understanding of food and skills from being immersed in value-based food systems, and confidence in peer-teaching. Experiencing and becoming part of the food value chain through FND built food literacy, shifted values, and transformed students into food citizens.

First two paragraphs are written in a narrative style—not an academic format. It is more appropriate for a book chapter than an academic journal.

Thank you for your comment. While qualitative Research (QL) is different than quantitative research (QN) and narration is more common we removed the introduction sections as per your and one other reviewer request.

The introduction could be improved in a few places including the final concluding paragraph. I’m not sure why “(and farm)” is inserted—seems tangential. The significance/relevance of the research topic should be more clearly articulated. This would help us better understand why this gap in research is important to address.

Thank you for your comment. We added two paragraphs at the end of the introduction to create more clarity on the significance related to this study. These two paragraphs highlight the significance of this study for both 1) nutrition students and their place in food systems work and 2) the role of university universities in general, and dining halls specifically, for food literacy experiences for students and the lack of data in nutrition students and their experiences in value chains.

We removed farm literacy – this term actually has meaning in our nutrition students’ experiences because of the lack of agriculture in their curriculum. Farm literacy stresses all that would need be known about agriculture to increase not only food literacy but also citizenship. Regardless, we followed your advice and removed farm literacy everywhere throughout.

“…the aim of this study was to evaluate pathways of food literacy and citizenship in university students, faculty, and staff experiencing eating at, volunteering for, and leading Food Next Door, a student-run, local food establishment, on a university campus.”

Thank you for your focused comment here and we agree, we did not do a good job in highlighting the pathways. Now we include a clearer purpose statement with secondary aim to explore pathways from food literacy to citizenship through the experiences in FND. See abstract, line 19, at end of intro 232-233.

‘Pathways’ we now include better in the two paragraphs at the end of the methods, starting on line 482 and ending 495. We include them as part of our triangulation and define them on line 493 and 494. In the result section, we introduce the qualitative results for which we also added a brief reference to pathways in line 753. In section 3.2.4. on line 1353 we added a paragraph describing the pathways from food literacy to citizenship followed by figure 3 which shows this progression using an abbreviated version of the themes and sub-themes of interview data from customers, volunteers and leads. We simplified this figure with less words in the cloud.

The research population isn’t clear: is it “students, faculty, staff” or “customers, volunteers, leads” (ln 219)

Thank you for your comment. We felt we described our participants in each section carefully but added more detail on the customers, where it lacked detail.

For the pilot we included students, faculty, and staff (this is shown in the abstract) and also in the method section (line 460). In the 3 samples, customers included mainly students but there were faculty and staff as well. We added this in line 462. Volunteers and Leads were all students, both undergrad and grad. We describe them under participants line 463-464.

And, what are "pathways"?

See above comment

Methods overall could be improved to reduce redundancies and to better clarify the timeline of the study. For example, what was the purpose of the pilot study? Where did interviews occur? An image / images of the farm/cafe would be helpful. I find the syntax awkward, which makes it difficult to read. It’s unclear if pilot results were used to design Phases 2 & 3.  There isn’t a clear link between DOI and methods / research design / analysis. It’s confusing as to whether FND actively educates and promotes food literacy or if it is a latent effect? What are the research questions or objectives guiding research design? Where participants tracked to see if there was an increase in their food literacy / citizenship?  If not, how do we know which “pathways” impact food literacy / citizenship? Without that, it is difficult to interpret results.

Thanks for specifying the areas we need more clarity. First of all, Figure 2 provides the cyclic visual of FND. We are also including another figure to show the actual station in both retail and residential dining for the reader to get a visual idea.

For interviews, they were held at the UCCS Farmhouse, which is now included on line 563.

For DOI, we provide more detail now (see section 2.2, lines from 387) to clarify that our intent was to study the experiences of FND in those eating at, volunteering at, and working for FND, which we refer to as early adopters using an exploratory design. We express that we did not determine diffusion rate as we did not study any other population group than early adopters. We recommend this for future. We focused on early adopters a priori, which is an approach based on Creswell 2003 that suspends the adoption of any theoretical structure prior to data collection and analysis. This comment here also pertains to further comments in the reviewer’s paragraph above, namely, ‘there isn’t a clear link between DOI and methods, research design, and analysis’. We however tried to accomplish this in a few areas (lines 553 in methods, analysis line 747), based on the reviewer’s request. As described above already, we included more on DOI in methods.

The comment on ‘pathways’ we explained previously. We can add here that we did not measure food literacy or citizenship in the small qualitative samples. This is our plan going forward to assess these constructs in larger samples sizes using surveys. The pathways from food literacy to citizenship were developed from this case study using an inductive process by Stake (see lines 479), and this process showed that FND built knowledge and skills (food literacy) first before individuals began to transform into food citizens. FND per se provides food literacy outreach to customers (see themes in customers) but in itself it built food literacy in nutrition students who were both volunteers and leads. So, to the comment above ‘its confusing as to whether FND actively educates and promotes food literacy or if it is a latent effect’, we really appreciated and adopted into the paragraph of pathways (3.2.4. lines 1360) where we now say: FND promoted food literacy in customers, while in volunteers and leads, food literacy was a latent effect of FND experiences. As for qualitative research methods used for this study, we did not aim for predictability or causation but our aim was to explore experiences from various groups being involved with FND. We show through the qualitative approach that pathways are possible in all three groups but we do not claim causation, as we were not interested in this but rather the essence of the experience and the variable progression from food literacy to citizenship.

The timeline of the study is expressed in different areas, first starting on line 446, 467, 475, 556 with reference to the years with years 1 in 2.4.2. and years 2 and 3 in 2.4.3.

For the pilot study we added more information starting on line 547, with the purpose of the pilot being to build the case of FND.

We are unclear what is meant with the syntax comment. We assume it refers to the italics, representing quotes. We inquired with the journal how to best use quotations and whether there were any directions for authors. The journal mentioned that we could use what we thought would meet our needs as long as we were consistent, and this is what we followed. We also introduced how we refer to quotations in the line 899 for the reader to understand the syntax/writing style.

Paragraph ln 331-339: Basically, this is reporting that those with higher food literacy scores were more likely to purchase and pay more for the food. I suspect it is more self-selection rather than causation…

Thank you for this comment. We are not claiming causation here. The aim of this study was exploratory and the pilot work was done to build the case for FND. We added a bit of information on the food literacy scores in the discussion, starting on line 1623 pertaining to food literacy opportunities in college and again in institutional procurement with respect to value chain-procured foods costing more but people also being willing to pay more for better food, especially if they are food literate see lines 1641-1660.

3.2.1, results need to be more directly linked to the research aims. I find much of the summary tangential—and frankly not that interesting. For example, the fact that customers return because the staff is friendly is inconsequential.

Thank you for your comment. We addressed our research aim and feel our results are linked with the research aim. In our opinion, the data of the customers, who were mainly students, show the importance of a friendly and inviting food experience. As highlighted by Malan, 2020, there are many students who struggle with college and the food transitions from home to independence. Creating a helpful and inspiring environment is critical for young people to trust and return to. We highlighted this in the paragraph on food literacy in university environments, see lines 1630-1635. 

3.2.2 & 3.2.3 are more aligned with research aims but could be more clearly organized

Thank you for your comment. These two areas pertaining to volunteers and leads. Through rearranging the discussion and removing most of the narratives into the findings (under these headings) we separated more clearly the finding from the discussion. These two areas are linked to the interview questions and the themes and sub-themes. We start each section with a summary of the themes, followed by the table showing themes and sub-themes, and narrative explaining these results by themes (by paragraphs). Figure 3 shows a visual of these themes and sub-themes. We added a summary paragraph at the end of the two sections and an overall quote from an FND Lead, summarizing the experience of FND. We hope this helps to elevate the experiences of the nutrition students and tie the information on the two section together better.

Much of the discussion belongs in findings—it is the summary of data analysis.  The discussion should focus on higher level and key findings rather than a more narrative and specific summary of results. It should also link key findings to the literature in general, which starts to happen in Ln 586 onwards.

We addressed this overall with shifting much of the narratives from the discussion into the findings and focusing the discussion on key findings, limitations, recommendations, and future research. This section starts on line 1374.

Re: Limitations: a small sample is not necessarily a limitation, particularly with qualitative research. It would be better to give the audience an idea of how many people were involved with FND to get a sense of how the sample was representative.

Thank you for this comment. We added some numbers under the limitation and sample size see lines 1733.

Re: the food literacy survey—how were reliability and validity ensured? What about construct validity?

Thank you for this comment. We did not test construct validity. As mentioned in the methods, see lines 415-418, the survey was from a different study which unfortunately remained unpublished. Due to the fact that food literacy is such a complex and evolving topic and our research was shifting to focus on the experiences of FND using qualitative research, we used the food literacy survey we had available but knew that it was only comprehensive enough to capture data for a pilot study. Our intent is to go back to this area and re-evaluate our survey and expand to include what has only recently been developed and is continuously being developed.

Finally, the fact that the author lives on a farm is not a bias—the study wasn’t about farms but about literacy. The two are not the same.

Ok great! Thanks for this perspective. We removed it.

Recommendations: where are they? This section is vague. Suggestion to re-write with specific actionable recommendations for practice, research, and theory. Or delete.

Thank you for this comment. We shortened this area. The first paragraph makes actionable recommendations regarding the inclusion of sustainability and learning opportunities in institutions, especially focused on health professions and nutrition. In the second paragraph I highlight the work that is currently being done and how our research can provide some can be used data in the development of curriculum and/or competencies. See lines starting at 1748.

At the end of the paper, I am left wondering what ARE the pathways to food literacy and citizenship?

Thank you for your thoughtfulness in the review. We hope that it is now clearer.

Conclusion: The authors haven’t clearly provided evidence to support the statement Ln 708-709.  Conclusion should be improved.

Thank you for your thoroughness! We completely modified the conclusion. See line 2101.

In reading Appendix A, I suggest organizing findings around the research questions. These should also clearly link to the DOI in the methods section.

We believe that our research aim now highlights clearly the selection of interview questions in Appendix A and we referred to the appendix in the methods. Likewise, the results from Appendix B are referred in the respective finding sections. Again, our DOI was not a full scope of DOI but we chose to remain exploratory and not to follow a structure. This was also given because of the early adopters we chose to study.

In general, while the narrative style is accessible, it is not appropriate for an academic journal. It lessens the authors’ credibility by implying subtle biased opinion throughout.

Thanks very much for this. As author and FND pioneer, it is hard to write about such a rich experience in a more technical abstract way. As Pauly asserts, when faculty immerse themselves in experiential work with their students it is not only life changing for the students but also for the faculty. I am especially grateful for your objectivity and thorough review of this work. We hope together as a writing team that we have addressed most, if not all, areas of concern, especially the subtle biased opinions.

Specific items:

  • Abstract: what does “intrinsic” mean, i.e. intrinsic case study? Why is this representative of an intrinsic case study as opposed to, say, a critical case study?

Thank you for this question. Delineating our study as an intrinsic case study was primarily because FND itself was of primary interest to us. Our intention was not to execute a mixed methods empirical analysis of a case, to link the results to any theoretical positioning or argument, or to compare and contrast FND with other cases.  It was to focus on this case specifically by performing an in-depth exploration of participants’ experiences and perspectives unique to this situation. If you want to, you can cite Creswell, 2013 and Yazan 2015.

  •  
  • Numerals 1-10 should generally be written out. Exceptions are when it is with a larger number such as Ln 65-66

We addressed this throughout.

  • Ln 71-71: Citation? Added
  • Ln 75: (Wicked!) functions as both a reference to wicked problems as well as a value-laden term…not appropriate for an academic paper. We changed to complex. Wicked problems are also used in the academic literature but removed it. They come from climate science and were first used, I believe, by the Stockholm Center for Resilience. Wicked problems are more complex than complex problems…
  • Ln 81: Citation for the assumption that professionals have high food literacy? Added
  • Ln 120-121 is distracting; omit ok we removed this
  • Ln 124-125: Avoid single sentence paragraphs. ok
  • Paragraph Ln 161-186 is really long; is all this background information relevant and needed for this paper?
  • Thank you for you comment. We shortened this paragraph a bit but we do feel it is necessary to share what initiated FND so that it may be replicated elsewhere. These programs don’t just start. They require relationship building and community engagement. The paragraph also seems important because food literacy speaks so quickly about agriculture but there needs to be more discernment in terms of why nutrition students should spend some time in agriculture. It is also important these experiences, especially if they are in curricula do not remain electives for students. If they become core courses or are part of required course sequence they have a chance to become institutionalized which would mean they can be sustained. Finally, the flying carrot is mentioned here because of FND’s start with a bank of recipes and experiences cooking and making menus using locally grown food and food literacy using health and sustainability messaging. In the Lead themes some of the experiences were related to these recipes so we thought it would be important to have this background.
  • Ln 195: Citations??? We removed this sentence
  • Why was purposive sampling used (besides to illuminate understanding)—what were inclusion/exclusion criteria? We edited and provided a bit more detail on sampling. Inclusion criteria were only focused on customers and volunteers returning and FND leads had to be graduate assistants. This is highlighted in 459-466.
  • 1 It says there were four sources for selecting participants but three are given Yes, we apologize. We changed this in line 459.
  • Ln 261: Not necessary to state that a conference presentation/paper remains unpublished Ok
  • Paragraph Ln 263-270: Some of this should be in findings, for example, how many participants and dates of data collection Thank you for this. We moved some of the paragraph to line 763.
  • Table 1: Students above/below C—are data in the correct column? We removed the scores from the table as they did not align. They are now in the narrative on line 769.
  • Table 2: What is the relevance to the study? Increased sales isn’t indicative of increased literacy. Thank you for this comment. We included this because FND was not going to happen without fiscal sense. And sales are a highly objective indicator in dining operations to show success. While it is not food literacy per se, if sales increase it is a good sign of DOI and ultimately of greater food literacy dissemination. We included some wording on line 551-555.
  • Ln 346-347: Not necessary to include the sentence beginning “Verbatim quotes”; omit. Ok we removed these but they are typical in qualitative research. Verbatim means that they are true to the participants. We assume that it is logical without it. Thank you.
  • Ln 507: And compost??? We removed this. It means that it includes everything from cradle to grave or farm to resource recovery, which is compost.
  • Ln 515-534: This quote is unusually long and is presented as raw and unanalyzed data in the discussion—paraphrase and move to results. We shortened this quote a bit and edited it. We also moved it out of the discussion into the findings section as you suggested. We kept the quote though because it is summative and provides and overall student-centered expression of the circular experience of FND.
  • Ln 600: What do the authors mean by “progression”? Perhaps “hierarchy” is a more appropriate term. We chose progression over hierarchy because progression indicates growth, advancement, evolution and could mean breakthrough. Hierarchy is less dynamic and refers more to grouping or a pyramid shape. FND and the progression from food literacy to citizenship, based on our QLR, is more dynamic and there is a possible threshold or certain experiences (we don’t know which ones without further research) that result in a transition to food citizenship. One of the student leads mentioned “a light bulb went off”, indicating a possible breakthrough or “aha” moment and this lead also mentioned it in the context of the customer (see lines 1323).
  • Ln 601: How can the authors claim a “deeper understanding” occurred? Authors need to more explicitly describe interview questions so we know how change was measured. Thanks for this question. QLR is about gaining deeper understanding of phenomena. But we appreciate your comment. The interview questions are shown in Appendix A. As main researcher, having triangulated the data sources, the experience would not be sufficiently explicated if I chose “leads had a greater understanding”. With deeper understanding we mean that they understood the issues because of their immersion within FND. Other words could be “profound” but the term larger, higher, or terrific (all synonyms of greater) do not express what happened in the leads. May I refer you to line 1313-1326 and the highlighted quotes “I started to know food differently” “actually being fully submerged in doing it”. There is other evidence in the finding sections that show food systems knowledge, problem solving, decision making, navigating. Through these types of experiences, the leads developed deeper understanding and what it means to bring food grown nearby to campus eaters. There were many systems to span for the leads and many people with whom they had to collaborate and communicate. The interview questions, pertaining to the overall impact of FND on how the think about and act around food and what was most important to them about FND were the questions where themes of food systems knowledge and advocacy came from, and this what we mean with this deeper understanding. We adopted your suggestion and now have the two interview questions listed on line 1595 and 1596 with more detailed explanation that led to this result. We also chose to dampen the expression by “leads seemed to have gained a deeper understanding”
  • Paragraph Ln 663-671: Suggestion to omit; not relevant. We removed this. Thank you

Reviewer 3 Report

The authors have study an interesting topic. But some modifications are needed.

The indroduction part, must be rewritten without to much

personal thoughts.  

Figure 2.to much text to the figure. Maybe put all this text to the manuscipt and improve the visibility of icons.

Table 1. Undergrad. Student. Without foul stop.

Author Response

The authors have study an interesting topic. But some modifications are needed.

The indroduction part, must be rewritten without to much

personal thoughts.  

Thank you for your comments. We removed the first two paragraphs and edited the introduction.

Figure 2.to much text to the figure. Maybe put all this text to the manuscript and improve the visibility of icons.

Thank you for your comment. We have modified the figure with reduced terms and more text in the finding. See figure 2. We also modified figure 3, which now has less text.  

Table 1. Undergrad. Student. Without foul stop.

Done

Round 2

Reviewer 2 Report

Really nice improvement of the paper. Two minor comments: 

  1. In regards to writing style: I agree fully that academic papers can have an accessible writing style where the author's (s') voice is clear--in fact, the researcher should be acknowledged as part of the research. I want to make a distinction of this against what I termed "narrative". Narrative has an undertone of bias due to use of value-laden words--this is unnecessarily distracting to the audience and cheapens the rigor of the research (whether qual, quant or mixed). Academic writing can be well-written, in an accessible style without being "narrative." In any case, the writing style of the paper is much improved--and still retains the authors' voice.   
  2. Limitations and Conclusion could be improved but not necessary.  As written the limitations don't cover typical topics of generalizability, missing data, construct validity, etc. And, conclusion summarizes study findings rather than elevating up (however, that is done in the discussion, so I think it's fine).